# Living Well as a Muslim through the Pandemic Era—A Qualitative Study in Japan

**DOI:** 10.3390/ijerph19106020

**Published:** 2022-05-15

**Authors:** Ishtiaq Ahmad, Gaku Masuda, Sugishita Tomohiko, Chaudhry Ahmed Shabbir

**Affiliations:** 1Department of Global Health Research, Graduate School of Medicine, Juntendo University, Tokyo 113-8421, Japan; 2The Section of Global Health, Department of Hygiene and Public Health, Tokyo Women’s Medical University, Tokyo 162-8666, Japan; masuda.gaku@twmu.ac.jp (G.M.); sugishita.tomohiko@twmu.ac.jp (S.T.); 3Department of Medical Quality and Safety Management, Faculty of Medicine, Osaka Metropolitan University, Osaka 558-8585, Japan; chahmed1622@gmail.com

**Keywords:** migrants, Islam, COVID-19 pandemic, public health

## Abstract

This study explored the living situations, financial conditions, religious obligations, and social distancing of Muslims during the COVID-19 pandemic. In total, 28 Muslim community members living in the Kanto region were recruited; 18 of them were included in in-depth qualitative interviews and 10 in two focus group interviews. The snowball method was used, and the questionnaires were divided into four themes. The audio/video interviews were conducted via Zoom, and NAVIO was used to analyse the data thematically. The major Muslim events were cancelled, and the recommended physical distancing was maintained even during the prayers at home and in the mosques. The Japanese government’s financial support to each person was a beneficial step towards social protection, which was highlighted and praised by every single participant. Regardless of religious obligations, the closing of all major mosques in Tokyo demonstrates to the Japanese community how Muslims are serious about adhering to the public health guidelines during the pandemic. This study highlights that the pandemic has affected the religious patterns and behaviour of Muslims from inclusive to exclusive in a community, and recounts the significance of religious commitments.

## 1. Introduction

Governments around the world have mobilised resources to combat the COVID-19 pandemic and non-pharmaceutical interventions have been introduced, including staying at home and closing all unimportant businesses. In April 2020, a state of emergency was declared in Tokyo and other prefectures in Japan, although it was not a strict lockdown that restricted the activities of the population. During this time, infection control measures were taken by religious authorities, including those representing Buddhism, Shintoism, Islam, Christianity, and others [1,2,3]. In many places around the world, restrictions were imposed on people’s gatherings designed to fulfil their religious needs [4,5]. In this pandemic situation, the Islamic community faced many problems in performing their religious activities, and lockdown measures severely restricted movement within and across countries, including the closure of schools, universities, borders, etc. [6,7]. Islamists, on the other hand, believe that practising religion (praying together) not only strengthens the connection to God but also to each other. Therefore, communal prayer meetings are a particularly important part of religious life.

Islam is often referred to as the religion of prayers. In the Quran, the words “Perform prayer (five times a day) and give zakat (a certain type of property or annual payment according to Islamic law that is used for charitable and religious purposes as one of the five pillars of Islam)” are repeated many times [8]. Faithful performance of this divine command is an act of man’s gratitude to Allah (SWT) for his blessing and an expression of boundless submission to His will. With the appearance of communicable diseases, the Islamic community has become aware of its responsibility. During the plague, Prophet Muhammad (PBUM) advised Muslims to restrict all mobility, including travel, visiting each other, and avoiding the sick. Quarantine was also recommended by Prophet Muhammad (PBUH) to prevent the spread of epidemics. The Sunnah emphasises the following, “If you hear that there is a plague in a land, do not enter it; and if it (the plague) visits a land while you are in it, do not go out of it” (Sahih al-Bukhari, 2005b). Prophet Muhammad (PBUH) thus laid down strategies that are still used today by public health organisations. The rules propagated by Islamic health sciences for dealing with infectious diseases and thus maintaining cleanliness and hygiene were far ahead of the Prophet Muhammad’s time.

The Muslim population represents one of the largest immigrant and religious groups in Japan. It is estimated that the number of foreign Muslims was about 59,000 in 2004 and 100,000 in 2010 [9]. Muslims in Japan have diverse backgrounds, such as nationality, ethnicity, culture, lifestyle and dress, and they adhere to a variety of customs and traditions, including prayers and fasting. As a result, people who have never met or interacted with Muslims may have stereotypes and misconceptions about them [9]. Indonesia, Pakistan, Bangladesh, Malaysia, Iran, and Turkey are among the main countries of origin in Southeast Asia and South Asia. Although the majority of Muslims in Japan live in the three major metropolitan areas (Greater Tokyo, Chukyo, and Kinki), the Muslim network has continued to grow throughout the country. There is a growing concern that the Muslim community in Japan is particularly vulnerable to COVID-19 because of religious obligations, such as praying in a large group of people, and because they follow social norms, such as sharing the same utensils at meals or intimate social behaviours, such as the blessing kiss on the forehead and hand of elderly family members, which are potential risk factors for the spread of COVID-19. This makes it difficult to isolate or distance oneself socially, which increases the risk of COVID-19 infection. In this way, COVID-19 social distancing and self-isolation measures are both unusual and ridiculous in the eyes of Muslims. However, this can be interpreted as ‘othering’ and victim-blaming. The need to disentangle these different factors and gain a better understanding of how Muslim migrants reacted during the pandemic is also crucial for the Japanese community to gain a better understanding of their perceptions.

### Objective

This study aims to present the attitudes of Muslim communities toward the influence of the COVID-19 pandemic, with high contagious factors in the context of religious practices. An important element of the article is to show the Muslim perception of the epidemic, how they responded to the directives about COVID-19 from the Japanese government, practising Islam during emergencies, and experiences of social distancing, social isolation, and faith.

## 2. Materials and Methods

### 2.1. Study Design and Sample

This study is part of an on-going project funded by the Japan Society for the Promotion of Science (JSPS KAKENHI 19H04354) research grant that focuses on Africa, the Middle East and Asia, the origins of large-scale migration and refugees, to improve the living conditions of people in this region as “global welfare”. This branch of the study aims to understand the background of the migration of overseas Muslims living in Japan, who, despite the harsh conditions, devise various ways to make a living in Japan during the pandemic. 

This qualitative study used validated semi-structured questionnaires for the in-depth interviews; the duration was from September 2021 to January 2022.

Participants *n* = 28 were recruited through the snowball sampling method [10] using social media, this approach allows the researcher to explore the study objectives in-depth, which is particularly useful in under-researched areas because a deeper interpretation of the phenomenon under investigation can be gained [11]. The knowledge gained through this ongoing project cannot be generalised as a quantitative study, although it can provide a better comprehensive picture of local in-depth phenomena.

Inclusion and Exclusion Criteria:Aged over 18 yearsEnglish/Japanese/Urdu/Hindi speakersLiving and working in the Kanto regionParticipants with severe sickness and not residing in the Kanto region were excluded.

### 2.2. Data Collection and Target Population

In the purposive sampling of the Muslims living in the Kanto region (Greater Tokyo), adults (aged 18–60 years) were requested to take part in the study. The potential participants were invited via social media. We also made a study website with the outline and objectives: https://harmoniamonde.wixsite.com/globalwelfare2021 (accessed on 10 September 2021). We used this website for the recruitment process and a better understanding of the participants. The information sheet and consent form were provided electronically to potential participants as soon as they registered for the study. To follow up on the study details and ensure agreement, phone calls were made the next day according to times convenient for participants. Regarding the focus groups, participants who agreed to participate were invited to join the scheduled zoom online focus group session. Participants gave their permission to record video/audio interviews, which lasted from 40 to 50 min. We asked participants to read the consent form, sign it (consent and recorded verbal consent), and send it back to us via email before they were interviewed. According to participants’ preferences, interviews were conducted in English, Hindi, or Urdu. Researchers transcribed the audio verbatim, after which they reviewed and anonymized the transcripts before importing them into NVIVO software for analysis and coding.

### 2.3. Data Analysis

To interpret the results of the study, the research team blind-coded the transcripts, then collectively identified the codes and derived overarching themes. One researcher transcribed the English interviews verbatim. When translating and transcribing the Urdu and Hindi interviews, the same person translated and transcribed simultaneously. We conducted a thematic analysis that involved the following steps:Familiarisation with the dataCollection of initial codesSearch for themesVerification of themesDefinition and naming of themesProduction of results

Using this framework, the researchers categorised the interview transcripts into potential themes. Data from the focus groups were used to further explore the initial themes and to add new themes to the framework. Overall, we reviewed the initial themes and sub-themes from both data sets and identified a few overarching themes and sub-themes.

## 3. Results

Most of the participants were employed; three were self-employed or engaged in part-time work. Table 1 shows that the age of the participants ranged from 18 to 60 years. Among the participants two were housewives, six part-time worker, four participants enrolled in university/school and the remaining participants had full-time jobs. The ethnicity of the participants was mainly described as Arab, Asian (Pakistani, Indian, Bangladeshi, Afghan) and European Muslims. Even after deliberate efforts to include the elderly, we were unable to include anyone older than sixty. The intention was to recruit members of the Muslim community of all ages through active and purposeful engagement. For this reason, we were unable to directly explore the perspectives of this older age group, although some participants spoke about the perspectives of older members of their families and community.

### 3.1. Qualitative Findings

Muslim communities in Japan are not a homogeneous group, as they are diverse in terms of ethnicity, language, background, cultures, and traditions. To represent the diversity of Muslim groups in the Kanto region, the study aimed to include participants from different Muslim backgrounds. In total, 28 participants took part in the study, 18 of whom participated in individual interviews and 10 in two focus group interviews. Four main themes and subthemes (Appendix A) are listed that help to understand the perception of the participant about risk and restriction-imposed experiences.

Living in Japan (Motivation to live abroad)Importance of public health directives (precautionary messages and information)Living and practising Islam in JapanFaith

### 3.2. Living in Japan (Motivation to Live Abroad)

All participants, including the focus group, spoke in the context of economic needs and good job opportunities as the main reason for living abroad (Japan), and of low wages, not being eligible for financial assistance and livelihood responsibilities as the main reason for leaving their country. Participants cited concerns, fears, and potential barriers to financial independence during the ongoing pandemic.


*Participant 1: “I think the number of jobs and opportunities for foreigners, Japan has a lot of job opportunities. If I talk about my country, there are no jobs but in Japan, even housewives can also find a job.”*



*Participant 3: “…there are two reasons, I think. First, we all belong to the professional background, so we came here to enhance our skills, secondly to become economically stable.”*



*Participant 4: “I was interested in Japanese manga and Japanese animation, so I wanted to come and live in Japan. I also wanted to explore their culture, and language and to have a better life.”*



*Participant 5: “In my case, I left my country to get a good life, job and education, that’s why I come to Japan.”*



*Participant 22: “It was very hard to leave my family back home during Corona but I have to do it…as I fear of expire my visas”*


### 3.3. Importance of Public Health Directives (Precautionary Messages and Information)

In the beginning stages of the pandemic, most participants reported that they were unable to obtain adequate information and advice. A focus group participant mentioned that “I was diagnosed with COVID, and the local hospital advised me to stay at home because I was having mild symptoms, the everyday morning I received to online fill the health status, in the beginning, I couldn’t understand because all the messages were in the Japanese language”. Participants generally said, “I think it is best to provide information in English.” The main criticism of public health measures was their inconsistency, which some said made no sense. Some participants said public health policies contained contradictory messages, such as instructing people to stay 2 metres away from others in certain situations but not in others, such as in schools or on public transport. Public health interventions require more resources, such as offering masks for free or providing additional resources to support distancing in schools.


*Participant 6: “… providing facemask in the early days of the pandemic was a great step by the government… we were unable to find mask, that was a great step”*



*Participant 3: “Yeah, they did their best and their information was very good in conveying the message every city hall was doing its best. “*



*Participant 7: “I think there was one issue of information the COVID-19 information was all on the government websites and TV but people who can’t understand Japanese they can’t understand about the information so I think the government should distribute other big languages in form of a pamphlet or anything.”*



*Focus group participant 2: “….I received very good support from the Japanese government when I got infected with Corona; city hall provided me necessary food I asked for...”*


#### 3.3.1. Satisfaction with COVID-19 Vaccination and Health Care

Studies have found that the willingness to be vaccinated against COVID-19 was associated with higher trust in information from government sources. [12]. Public vaccination in Japan began in February 2021 with medical workers (about 4.7 million people), followed in April by the elderly (about 36 million people) in June 2021, and then by all ages in July 2021. The vaccination process in Japan was very slow as compared to other countries, but now they have vaccinated about 84.5% of the country’s population [13]. Regarding this question, one of the participants stated that “…*when I got an infection I got the vaccination appointment but it was very late as I was already infected when I received a letter from City Hall, most of the clinics were full it was hard to get an appointment for first shot”*.

There was a mixed response as mentioned above about getting vaccinated against COVID 19 regardless of all age groups.


*Participant 13: “… I think throughout the world Japan did take a lot of time. I think because they wanted to do their best and they chose the safest one like Pfizer. They are dedicated to their work, especially to the COVID-19 situation. They are properly arranged and organized”*



*Participant 12: “Yes! about the COVID-19 vaccine, I can say the Japanese government was not serious about it last year because all over the world people were getting vaccinated, but the government was not taking it seriously now from this year I have seen a rapid increase in the vaccination process and people are getting vaccinated.”*



*Participant 3: “In my thinking, the Japanese government only publicize such information in the Japanese language. I think they should adopt multiple languages because some of the people who come to Japan are not educated.”*



*Participant 5: “I appreciate the vaccine process according to global news; I think Japan vaccine was a little bit late. Yes, I feel like other countries were fast in terms of the vaccine.”*



*Participant 10: “Yeah, I think there was no negligence from the government about COVID-19 sops. They did their best even though many other countries were unable to do anything like they were not able to understand the situation. But the Japanese government did its fine and stop all regular stuff to cope with the situation. Yes, there was and is the difficulty of language as for us we know little bit Japanese, so it was not so difficult for us but for other people it was.”*



*Focus group participant 1:” I tested positive with Corona, I didn’t have Oximeter and thermometer, I requested to city hall using their online application and next day they sent me, also every day I received the call to report my health condition. The only problem I faced was to translate the Japanese language messages into English.”*


#### 3.3.2. Social Isolation and Barriers to Local Communication

Participants discussed their experiences of isolation and their narratives pointed out that belief in the seriousness of the virus influenced how social distancing measures were adhered to. The main problem arose when cut off from their community and friends; in addition, it was argued that the initial messages did not convince people about the severity of the virus.


*Participant 6: “I think we possibly need to communicate with the community members within those areas, especially in our Asian communities, many things we depend on each other including information and some other matters.”*



*Participant 4: “I’ve seen those people now; I feel that people if they’re not necessarily listening to the guidance, it’s just a case of you doing what you want to do. Very early on when we were all hearing a lot of young people are fine, it’s the older people that are being affected as if that was ok and we were supposed to be ok with older people being at risk. I was like ok I understand what you’re trying to say and like it’s not necessarily like the plague that’s wiping everyone out, but there’s still a significant risk and people were just kind of like ‘oh well if it’s not affecting me and my friends, I don’t have to change the way I am.”*



*Participant 14: “…. Well in the start we did follow. We followed the social distancing it took two years sometimes we forgot, even Japanese does follow it previously but now they are not, for example, in trains, all I can say is they follow sops to some extent but not much.”*



*Participant 2: “I think this is one part of the hard situation caused by COVID-19 that we can’t meet each other, difficult communication, umm yes impacted such as meet up my mates meet up with friends work remote difficult in communication with worker and clients. During the COVID-19 in Japan communication got more difficult.”*



*Participant 11: “The closer of mosques for all kinds of prayer was the worst decision because it was the usual channel for communication with community for Muslims which was absent during lockdown … I was lost.”*



*Participant 6: “We live in a Muslim community area where there are many masjids, there was no such big effect on my life because we were maintaining the social distancing whenever we go out for prayers. So, it was fine for me to follow the sops and to say the prayer. We were going to university every day and following sops.”*



*Participant 25: “I was very hard to communicate during the online classes as mostly I used to take on-site lectures also my research work not completed I don’t know I can graduate on time”*


#### 3.3.3. Financial Condition and Perceptions of Self—Cultural Challenges

In addition to their perceptions about the seriousness of COVID-19, participants expressed their opinions about the possibility of contracting the virus, as well as their perceptions of how the virus is transmitted. Some participants hadn’t paid attention to the virus until they were infected with it or their family or friends fell ill with it.


*Focus group participant 4: “I must say thanks to the Japanese government, they did support everyone. They give 100,000 YEN to everyone. They even set many categories to expect the money. Everybody got the benefit.”*



*Participant 3: “…Yeah Japanese government took the step and they provided everyone with 100,000 yen so it was a good step I can’t say it fulfil my all need but yeah it was fine to get myself together and cope with the situation. No, we didn’t get any support from Pakistan because we are a developing county and I belong to the industry sector if the industry sake whole-sale get shaken”*



*Participant 2: “The financial support from the government was good but not enough if a person loses his job, he can’t compensate his expenses with this low amount so all I can is it was good but not enough according to Japanese life standards. They also supported children”*



*Participant 8: “Alhamdulillah I’ve been. I think it’s my job that’s kind of like grounded me. Perhaps and this is the truth perhaps if I’d worked in a different field maybe I wouldn’t have been as firm and stricter as I am now.”*



*Participant No 27: “I want to fly back to my country but ticket are too much expensive, I’m afraid I can’t see my family.”*


#### 3.3.4. Social Distancing and Limitations of the Environment

There was considerable uncertainty about the possibility of self-isolation among participants. For instance, one subject discussed the difficulties of self-isolation as a mother of young children.


*Participant 1: “Well…huh…. l I do isolation, but I will also attach with my kids… in Islam, it’s not like that the real thing I have been travelling a lot in Tokyo but none of my kids got COVID-19. But if it ever did happen, I will never separate I will never hospitalize them.”*



*Participant 7: “in a small office place is challenging to make sure isolation…. I work from home.”*



*Focus group participant 5: “I have been very unproductive with my work at home, I was not used to do the things at home.”*


### 3.4. Living and Practising Islam in Japan during the Pandemic

#### 3.4.1. Assessing the Risk of COVID-19—Taking Precautions

Women and children can receive relevant education in some mosques [9], adult men are generally offered it in most mosques. Some mosques decided to cancel Friday prayers and Qur’an classes, as well as restricting entry to buildings, during the early stages of the outbreak. Participants thought social distancing and isolation measures were important for reducing the risk of virus transmission, but their experiences showed that sticking to them was difficult for them as well as others. Droplets inhaled through the air ‘easily’ spread and transmitted the virus, according to participants. Participants were advised to practice good hygiene and maintain social distancing.


*Participant 9: “…The closure of the mosques was another important cue or sign as to the seriousness of the virus, identified as important by participants because the mosques are central to community worship that is a key part of Muslim community life and consequently mosques are rarely if ever closed.”*



*Participant 3: “Islam already taught us about such diseases our beloved prophet Harazt Muhammad says that ‘if such disease happens you are not allowed to leave the city or any place you are in’. So we already knew about the action. We believe if any person gets expire it’s because God wants him so if any accident happens or any other thing which causes his death it’s natural. So, it believes death is from Allah. If any person in my family gets this infection, I will attach one person with him to take care of him like shower food and stuff and with the precautions like gloves mask etc. because in Islam we don’t leave any person behind.”*



*Participant 12: “Yes there was the issue of our prayers like they suddenly stopped all the prayers in Masjids, so it was a difficult time but on the other hand, it was for the benefit of us. We Muslims did our best to follow the rules.”*



*Participant 5: “I know a few people who have had the corona at first they just thought it was nothing, it didn’t show, they didn’t have the temperature but they had like sort of a cough or whatever but they worked through that because it’s just that ‘oh I need to work through that and just carry on’ but it was only until they’d get tested and then they go ‘oh yes I’ve got it but you would have never thought’.”*



*Focus group participant 2: “I think closing down the Mosques was a big message for the community that we are experiencing serious disease yeah so you know Mosque can never be closed just if not this, coronavirus.”*



*Focus group participant 4: “The fact that lockdown took place during the month of Ramadan and the celebration of Eid where members of the Muslim community would usually break the fast in the congregation, was a particular challenge, with one participant reporting that whilst they had stayed at home, they had heard of other households meeting up.”*



*Focus group participant 4: “As Muslims, we shake hands and give hugs to each other and thus it’s our nature …. we can’t just give it up just easily and this is a big challenge for all of us as Muslims. We want to protect ourselves as a community and protect others of course I can see a lot of Muslims just following these measures.”*


#### 3.4.2. Prayer Cancellation (including the Holy Month Ramadan and Friday) and the Practice of Cleanliness

Members of the Muslim community must maintain cleanliness before regularly scheduled prayers, the Salaat (Namaz), which takes place throughout the day. Religious teachings and practices play an important role in promoting hygiene practices, including frequent hand-washing. Several participants stated that all mosques in the Kanto region have taken several measures to prevent the outbreak, such as refraining from mass prayers and closing mosques when the pandemic broke out, cancelling major events during Ramadan, and moving to the online medium, ensuring proper ventilation inside mosques, and maintaining a safe distance when attending personal prayers [9]. Furthermore, mosques, including those serving members of the Muslim community, have provided support to those affected by COVID-19, as they have done during natural disasters in the past. However, many Japanese are unaware that mosques can be used for shelter and home if needed during any emergency. Therefore, the above measures can compensate for their lack of knowledge and reduce the risk of COVID-19.


*Participant 3: “Yes it affected our prayers, I often pray with my wife at home for Juma prayer especially we gather some other people like my neighbours, and we offer prayer in the gathering but not every prayer. During prayer, we always make sure to keep a social distance. Yes, it affected our Ramadan life, we could not be able to gather in the mosque for Iftari but we tried to gather to keep our mosque alive, we reduce the person to 15 people, so only 10 to 15 people were allowed to come to the mosque to pray Juma prayer and to do Iftari.”*



*Participant 6: “Yes you know the COVID-19 impacted our prayer very much. When the government impose an emergency only a few peoples we coming to the masjid for prayer only the imam or some peoples. We stopped many religious activities due to corona and SOPS from the government. Five-time prayers are mandatory but if you have any symptoms our Ulma’s were advising us not to come to the masjid and stay home and pray there.”*



*Participant 12: “I heard of people meeting up on Eid … like different households so that’s obviously up to them. Personally, I didn’t I didn’t meet up with anyone.”*



*Participant 13: “One thing I do like is the emphasis on cleanliness you know we have that within our communities anyway you know we wash 5 times a day for prayer.”*


#### 3.4.3. A Sense of Responsibility (Duty)

For a Muslim, a strong belief is in the responsibility of following the saying of the Prophet Muhammad (PBUH). Almost all the participants mentioned that the sense of responsibility reinforced the need to follow state guidelines about COVID-19. Moreover, participants emphasised that they wanted to present themselves as responsible foreign Muslims.


*Participant 3: “We believe trusting in Allah during troubled times is essential, but opposing sanitary rules is a bad idea since these are necessary to stop viruses from spreading.”*



*Participant 5: “As Muslims, we have a responsibility to God, to our relatives’ other peoples and ourselves. So, it’s one of our responsibilities to follow the government guidelines and to act accordingly.”*



*Participant 2: “as most of us depend upon on social media, so it was sort of reflecting what the government was telling us and, we knew that COVID-19 was spreading quite rapidly in other countries. So, we were unsure what the severity of coronavirus would be and but knew some restriction to follow as responsible.”*



*Participant 13: “my bother in Japan doesn’t get her information from the news, I don’t know the conversations he’s having with his friends, and I don’t know whether they are acknowledging the seriousness of this.”*


#### 3.4.4. Provision of Support (Donations)

Having support outside the home was also important for isolation and self-shielding. Participants described early challenges where there were no systems in place to help vulnerable individuals to isolate themselves, while others described how members of the community delivered essentials to those in need so they could protect themselves.


*Participant 4: “I think there was an increase in the donation, people lost their jobs but the government did not support them but we have a zakat system so we help poor people but in Japan, I think there are no poor people like my county.”*



*Participant 5: “Yes donations got increased we also collected the money from Japan and sent it to Bangladesh we were trying to give money to people who don’t have a job in my country.”*


### 3.5. Faith and Adherence

As the participants’ faith influenced their beliefs, actions to minimise the risk of getting themselves and others infected were in turn influenced by their beliefs. It was important for them to refer to key religious texts that offered instructions for action to support the instruction to distance socially and isolate them when needed.


*Participant 9: “…Islam mandates you to take necessary precautions and at the time of the Prophet Muhammad there was a plague epidemic, and he guided his companions to take precautions and to make sure the essential practised isolation.”*



*Participant 13: “Religious texts were important in the context of the closure of the mosques and disruption of usual religious practices such as breaking fast in congregation at the end of Ramadan. Participants described how they felt these changes regarding their faith and in so doing, did not feel they were making unacceptable compromises, which might have influenced their adherence.”*



*Participant 4: “The correct understanding is that this situation allows you to pray at home and your prayers will still be acceptable permissible and you are not meant to risk yourself for any such situations.”*


## 4. Discussion

This study found that the Muslims in Japan are facing numerous problems since the beginning of the pandemic. During this period, there has been an increased social distancing between the communities. In other words, this distancing has nothing to do with the virus prevention but a lack of social contact within the Muslim community, which is leading to loneliness and a sense of alienation. This causes depression-like symptoms, along with constant concern for one’s own and one’s loved ones’ health [14]. Additionally, participants were concerned about their economic conditions, and those with poorly protected jobs with low wages. There is a great need for paying more attention to low-paying jobs and part-time employment. Among the three part-time workers, one stated that “I got Corona and my company didn’t pay me for the days I quarantined at home”. Furthermore, economic hardship was identified as a primary factor driving greater exposure to the virus, as people fear losing their jobs if they cannot get financial support. Psychological instability is exacerbated by the economic stress associated with the COVID-19 crisis, such as rising poverty and unemployment [15,16].

It is not easy for most of the local people to understand what kind of infection control measures are implemented in mosques because they are unfamiliar with Islam [17]. The spread of misinformation is polarizing public debate on COVID-19-related topics and Islam, raising the possibility of conflict in religion, and social cohesion [18]. Having access to this information will help people to gain a better understanding of unfamiliar Islamic ways of hygiene and contribute to a multicultural society. We found that perceived consequences for others were not a motivator for hygiene behaviours in a recent survey conducted in the United States during the early stages of COVID-19 [19].

Participants in this study considered adherence to social distancing rules as a religious duty that compels them to take the virus seriously. Ramadan, for example, celebrates the fasting month every year, in which healthy adult Muslims fast from dawn to dusk every day [20]. A recent study has shown that daily intermittent fasting during Ramadan has beneficial effects on COVID-19 [21]. There is evidence that intermittent fasting during Ramadan may have a limited beneficial effect on immunity due to sleep habits during the blessed month of Ramadan. However, several COVID-19 studies have shown that total sleep time decreases by about 1 h during Ramadan nights, while daytime sleepiness increases as people stay at home or even work from home [22,23]. Islam emphasizes praying and having faith, as shown in the holy Quran many times as well as during the holy month when Muslim prayers tend to increase more than usual.

According to our study, the information provided by different Japanese authorities related to COVID-19 was unsatisfactory because most of it was in Japanese. Participants have shown their concerns which is very crucial, as the credibility of the information provided by the authorities can be called into question if it is not understandable [24]. This is because decision-making requires the integration of information from different sources which increases the relative reliability [25]. The lack of relevant information in an inappropriate format, not including community languages, was identified as a problem in our research. Recent research on youth adherence to COVID-19 measures endorses disseminating public health messages through trusted resources [26]. A recent survey found that a significant proportion of the population received conflicting information that the coronavirus is an artificially modified virus; this kind of information may lead to depression [27]. This is consistent with existing research showing that exposure to inappropriate and risk-increasing messages causes stress, worry and anxiety in the public [28].

According to Murphy et al., compliance with COVID-19 social distancing requirements is influenced by demographic, instrumental and normative factors. According to these authors, a study on Australians shows that they comply not out of concern for themselves or others, but out of a sense of responsibility to help the authorities [2]. Risk perceptions and protective health behaviours in times of a pandemic have been linked in several studies. A 2010 study on protective behaviour during a pandemic [29] and a telephone survey in Italy found that perceptions of the severity of the illness and the risk of contracting the virus were related to adherence to some recommended behaviours [30]. An earlier study on beliefs and attitudes found perceptions of vulnerability to COVID-19. The perceived risk of contracting COVID-19 and trust in governments were less important predictors of taking adhering to protection measures than beliefs. The health protection measures would be effective in preventing COVID-19 with handwashing, mask-wearing, and physical distancing reducing the incidence of COVID-19 [31,32].

According to results from of our study, observing the social distancing rules was undoubtedly challenging, especially during the holy month of Ramadan in which breaking the fast together was not included, as usually, most participants found these deviations from normative practices quite acceptable and coherent with their religious teachings. Several studies have shown that COVID-19 restrictions have a significant negative impact on social life and well-being [27,33]. There is evidence that obligations, such as obedience to authority figures, play a role in breaking COVID-19 restrictions and creating a sense of duty from an early age. Our findings do not support the claim that Muslims ignore the advice and congregate in large groups. This can be interpreted as extraneous and as victim-blaming, especially in social media and the media in general.

The results of this study contribute to the emerging field of research on the well-being of Muslim migrants, including changes in their religious practices in the context of COVID-19. Participants were aware of the varying length of the recovery process from the pandemic and have the opportunity to adjust their workspaces in accordance with continuing social distancing regulations, and they may have the option to attend religious services from workplaces. Participants were also concerned about their health problems and chronic conditions such as diabetes and obesity. Because of the COVID-19 restrictions which limit the daily physical activity, there can be a risk of developing diseases such as CVDs and diabetes [34,35]. In terms of improving physical activity [36], mental health [37], and work productivity [38], there is a wealth of published research available. In addition, our study shows that clear and effective public health interventions are needed to improve well-being and facilitate health care.

### Strengths and Limitations

This is the first study to focus on how members of Japan’s Muslim community have dealt with the COVID-19 pandemic and how it has affected their lives. It offers suggestions for how information and communication could be improved in the future to ensure risk reduction information can be provided to the communities. Interviews were conducted by interviewers from the Muslim community, which facilitated the discussion and allowed the participants to use terminology associated with Muslim culture.

Studies conducted through online facilities always carry some uncertainty about data validity. Nonetheless, online surveys offer many advantages, including protecting the identities of participants, sexual orientation, gender base diversity and demographics [3]. This study only included the experiences of those who wished to participate; our findings may not reflect the experiences of all Muslim communities. We had some challenges in engaging older people in our research. Further research is needed to capture the experiences of elderly members of the Muslim community in Japan. Furthermore, this study was conducted at the height of the pandemic; people’s perceptions of risk may have changed over time.

## 5. Conclusions

Understandable public health messages to the public are essential since there is the potential for misinterpretation. The study highlighted that reliance on Allah (God) in troubled times is an important part of participants’ religion, but rejection of hygiene rules stems from misunderstanding, as these are necessary to prevent the spread of diseases. Moreover, practising religion was not a source of spreading infections, but not going to the mosque led to people being isolated from their community. Effective public health interventions that improve physical activity, mental health, and productivity are needed to improve the well-being of people who are working at home due to the pandemic.

## Figures and Tables

**Table 1 ijerph-19-06020-t001:** Participant characteristics.

	Frequency
**Age**	
18–30	8
31–40	13
41–60	7
**Gender**	
Male	18
Female	10
**Material status**	
Married	20
Unmarried	7
Widow	1
**Education**	
High School	5
Graduation	14
Master	5
PhD	4
**Occupation**	
Part-time	6
Restaurant worker	4
IT	1
Self-employed	3
Engineer	2
Factory worker	4
House wife	2
Professor	1
Researcher	1
Student	4

## Data Availability

Since the datasets generated and/or analysed during this study contain information that could compromise the confidentiality and anonymity of the participants, they are not publicly available but can be obtained from the corresponding author (limited) upon reasonable request.

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
