# Peer review of "Living Well as a Muslim through the Pandemic Era—A Qualitative Study in Japan"

_ijerph, 2022, doi:10.3390/ijerph19106020_

Round 1

Reviewer 1 Report

Please see the file attached.

Author Response

Thank you so much for your valuable comments. Please see the attached file. I would be happy to provide you with any additional information if you need it.

Reviewer 2 Report

Thank you for this paper. I have only minor suggestions as overall I found the rationale for the paper very acceptable and its content of interest for this and future pandemics. Suggestions include:

a) Line 41 - not just religious activities - lockdown measures severely restricted movement within/ across countries including closure of schools, universities, borders, etc. Some of these across Africa and other Asian countries are described in: 

Ogunleye OO et al. Response to the Novel Corona Virus (COVID-19) Pandemic Across Africa: Successes, Challenges, and Implications for the Future. Front Pharmacol. 2020;11:1205 and Godman B et al. Rapid Assessment of Price Instability and Paucity of Medicines and Protection for COVID-19 Across Asia: Findings and Public Health Implications for the Future. Front Public Health. 2020;8:585832

b) Methodology

i) Why 20 (? reaching possible saturation)?; how were the 20 selected - you talk about snowballing - especially since you mention in the Discussion about the different worshiping practices between men and women with going to the mosque, etc. (I will come back to this)? 

ii) How were the topics selected for the single interviews and focus group - was this a validated questionnaire (building on lines 97-98)? 

iii) Line 113 - Why only the Kanto region out of the 3 main ones? How representative is this of the Muslim population in Japan - good to expand on this to add robustness/ justification to the results and conclusion?

c) Lines 167 - 168 (and linked to the Methodology) - have to hunt in the methodology section that a mixed approach used. Why was this - need to justify this and the split between the two methods in terms of overall interview numbers

d) Line 224 - you state 'now they have vaccinated about 84.5% of the country’s population' - what date was this and dies this refer to the first vaccination or boosters, etc.? Good to clarify this.

e) Lines 332 - 334 - part of this should go into the Introduction to give more background to the paper. In addition - as mentioned above - were there any differences in the responses to men and women given the differences in worshiping styles? If not - why not?

f) Lines 525 - other potential references include: Talic S et al. Effectiveness of public health measures in reducing the incidence of covid-19, SARS-CoV-2 transmission, and covid-19 mortality: systematic review and meta-analysis. Bmj. 2021;375:e068302; Ng Y et al. Evaluation of the Effectiveness of Surveillance and Containment Measures for the First 100 Patients with COVID-19 in Singapore - January 2-February 29, 2020. MMWR. 2020;69(11):307-11; Girum T et al. Optimal strategies for COVID-19 prevention from global evidence achieved through social distancing, stay at home, travel restriction and lockdown: a systematic review. Arch Public Health. 2021;79(1):150

Author Response

Thank you so much for your valuable comments. Please see the attached files
